# Physics-empowered Molecular Representation Learning

## Abstract

Estimating the energetic properties of molecular systems is a critical task in material design. With the trade-off between accuracy and computational cost, various methods have been used to predict the energy of materials, including recent neural-net-based models. However, most existing neural-net models are context-free (physics-ignoring) black-box models, limiting their applications to predict energy only within the distribution of the training set and thus preventing from being applied to the real practice of molecular design. Inspired by the physical mechanism of the interatomic potential, we propose a physics-driven energy prediction model using a Transformer. Our model is trained not only on the energy regression in the training set, but also with conditions inspired by physical insights and self-supervision based on Masked Atomic Modeling, making it adaptable to the optimization of molecular structure beyond the range observed during training, taking a step towards realizable molecular structure optimization.

## 1 Introduction

Material simulation is a vast research field that spans understanding material's optimal structure, simulating microscopic dynamics depending on time, temperature, and pressure beyond the experimental resolution, and reducing trial-error loops in designing new materials. The foundation of this simulation is defining the energy at the atomic level considering interactions between numerous atoms, so-called many-body problem. Advances in theory and computational capability have led to higher predictability of energy with greater accuracy than ever before. Despite the tremendous advances, however, many-body interactions between atoms are exponentially complex, so increasing raw computing power alone has fundamental limitations. Thus, it is unavoidable to put an effort into reducing computational cost, as well as improving the prediction accuracy (*e.g.*, quantum mechanics), which has been a grand challenge in computational material simulations.

Quantum mechanical electron structure simulation is aligned with the direction to enhance accuracy. Specifically, Density Functional Theory (DFT; Kohn & Sham (1965); Parr (1980)) is one of the most successful methods in terms of accuracy, describing and predicting the structure of a material, dynamics based on temperature, phase change, and chemical reaction. However, the number of atoms that DFT can practically handle is limited to a few thousand atoms. Thus, it is complicated to compute the dynamics at a larger scale with DFT, which demands an alternative method to scale up.

Another direction is to reduce the computational cost by expediting computation through approximation while considering many-body interactions. For atoms of much larger scale, so-called classical force field approximations Harrison et al. (2018) have been developed. Various classical force field potentials have been proposed, sharing a standard scheme: formulating the term-by-term energy equation using chemical intuition and performing parameterization for the targeting system. This approach also has drawbacks. First, building a classical force field requires extensive human effort to define energy terms by understanding the target system and parameterization. Also, the potential is described in the bonding terms of the material, so its applicability is restricted to non-reactive materials within similar phases, since the parameter should be changed depending on the chemical environment. Computationally heavier ReaxFF (Senftle et al., 2016; Gomzi et al., 2021) exists, but it focuses on chemical reactions that require parameters tailored to the specific reaction. The force field is challenged to secure accuracy despite a great deal of effort, and generalization to various chemical situations remains a challenging question. Nevertheless, it can mimic potentials with few parameters, since it is based on equations and thus scales better than quantum simulation, which requires energy optimization by self-consistent calculations.

Recently, machine learning has made an impact on the material simulation. Specifically, machine learning (ML) surrogate potentials, which predict the energies of given molecules, have drawn much attention from the scientific community as they provide an alternative solution to the aforementioned inevitable trade-off between cost and accuracy of physics-based models. Several state-of-the-art ML surrogate potentials achieved high accuracy close to that of quantum mechanics while requiring significantly less computational resources than traditional DFT-based methods.

One might argue that we can produce an infinite training set using DFT, and a fully data-driven approach would eventually work well on this problem, just like on other regression problems. However, considering the exponentially increasing complexity of this problem and limited resources, it is desirable to equip the surrogate potential model with domain knowledge from physics.

In this paper, we propose a hybrid approach that combines the powerful expressive power of Transformers (Vaswani et al., 2017) with classical force-field-style equations. In particular, our paper's main contributions are summarized as follows:

- We propose a *physics-empowered molecular representation learning* method, actually preserving the optimal structure instead of simply fitting the single energy value.
- Taking advantage of Transformers, our molecular representations are trained in a self-supervised manner by *Masked Atomic Modeling*, inspired by the approach of Masked Language Modeling.
- We conduct extensive experiments to evaluate the proposed model quantitatively and qualitatively, introducing several novel approaches to actually evaluate the model's ability to capture optimal structure of a molecule.

## 2 RELATED WORK

ML potentials can be categorized into three types based on model complexity and history: kernel-based descriptors, fixed atomic descriptors, and learnable descriptors using deep neural networks.

**Kernel-based Methods.** Kernel-regression-based potentials are mainly applied to a single atom or a few elemental species, where the kernel method is one of the lightest forms of ML. Gaussian Approximation Potential (GAP; Bartók et al. (2010)), Smooth Overlap of Atomic Potential (SOAP; Bartók et al. (2013)), and Spectral neighbor analysis potential (SNAP; Chen et al. (2017)) are representative examples. These models can be trained on a small amount of data, but it is difficult to be extended to chemically complex cases.

**Fixed descriptors.** Behler & Parrinello (2007) introduced an ML potential model that uses the atom-centered symmetry function (descriptor) to describe the local environment of each atom and passes each descriptor value to the simple feed-forward neural network to map the total energy. These descriptors (symmetry functions) process distance and angle information between paired atoms within a specific cutoff and produce a single value for each descriptor. Behler-Parinello Neural Network (BPNN; Behler & Parrinello (2007)) series are the representative practical examples that increase model complexity for high-dimensional Potential Energy Surface (PES) compared to previous kernel-based methods. BPNN was the realistic and the first attempt to decompose the total energy as a sum of each individual atom's energy. A fundamental limitation of this approach is that fixed descriptors are insufficient to cover complex spatial patterns (*e.g.*, ring structures, bond types, or chemical functional groups), limiting the knowledge transferability between different molecules. Also, the original symmetry function does not reflect the chemical environment outside the cutoff at all Kulichenko et al. (2021). Despite these limitations, it achieved accuracy that no previous classical force field reached. It has been shown to work for systems with many atoms in a dense system with a few species Behler (2015); Kulichenko et al. (2021).

**Deep Learning-based Models.** Recently, deep neural networks have been actively applied to construct surrogate potentials. Most models in this category allow the chemical environmental information can be transferred between atoms over a greater distance than traditional models, providing a higher degree of freedom. As a specific example, ANI (Smith et al., 2017) extended BPNN by modifying its angular function. Gilmer et al. (2017) proposed a Message Passing Neural Networks (MPNN), specialized in learning from graph-structured data by updating hidden node states by combining messages from adjacent nodes. Since then, various graph-based approaches (Schütt et al., 2018; Gasteiger et al., 2020; Unke & Meuwly, 2019) have been proposed. MPNNs significantly improved accuracy in molecule-related tasks on QM9 (Ruddigkeit et al., 2012; Reymond, 2015; Ramakrishnan et al., 2014), while the increased model capability nest a risk of overfitting (Hawkins,

2004; Zuo et al., 2020). Recently, the Transformer (Vaswani et al., 2017) is applied to this problem as well (Cho et al., 2021; Thölke & De Fabritiis, 2022), following its success on natural language processing Devlin et al. (2018) and computer vision (Dosovitskiy et al., 2021; Lu et al., 2019; Sun et al., 2019).

# 3 METHOD

## 3.1 PROBLEM DEFINITION AND NOTATIONS

Given a molecular structure graph $\mathcal{G} = (\mathcal{V}, \mathcal{E})$, where $\mathcal{V}$ is a set of $N$ atoms consisting the molecule and $\mathcal{E}$ is a set of bonds between a pair of atoms with direct interaction, we aim at a regression problem to estimate the energy $E_{\text{mol}} \in \mathbb{R}$ of the molecule. The total energy at the molecule level $E_{\text{mol}}$ is decomposed into the atomic level, denoted by $E_i$ for each atom $i = 1, ..., N$, where $E_{\text{mol}} = \sum_i E_i$. Each atom $i$ in the molecule is represented by its atomic number $z_i \in \mathbb{R}$, its position $\mathbf{p}_i \in \mathbb{R}^3$ in Cartesian coordinates, and electro-negativity $n_{z_i} \in \mathbb{R}$ of the atom type. We denote the pairwise L2 distance matrix $\mathbf{D} \in \mathbb{R}^{N \times N}$ between atoms, computed from $\{\mathbf{p}_i\}$. Here, the element $d_{i,j}$ is the radial distance between two atoms $i$ and $j$. Adjacency matrix that represents bond information of the molecule denoted by $\mathbf{A} \in \{0,1\}^{N \times N}$.

## 3.2 ATOM REPRESENTATIONS

Each atom first needs to be represented as a vector before we feed it into our Transformer-based model. Here, we describe how we represent each atom in detail.

**Atom-wise Representation.** Atom $i$ is embedded as a vector $\mathbf{x}_i^{(\text{self})} \in \mathbb{R}^d$ based on its type $z_i$ and its electro-negativity $n_{z_i}$:

$$\mathbf{x}_i^{(\text{self})} = [E(z_i); n_{z_i}], \tag{1}$$

where, $E$ is an embedding layer, and $[;]$ indicates concatenation.

**Radial Basis Functions.** Inspired by the localized orbitals in DFT, we start with a simple Gaussian basis to represent the relationship between two atoms. For a pair of two atoms $i$ and $j$ in the molecule, we assign $n_b$ basis functions following Unke & Meuwly (2019):

$$\psi_{i,j,k}(d_{i,j}) \equiv \varphi(d_{i,j}) \exp\left\{ -\beta_{z_i,k} \left( \exp(-d_{i,j}) - \mu_{z_i,k} \right)^2 \right\} \tag{2}$$

where $i = 1, ..., N$ is the center atom index, $j = 1, ..., N$ is a neighboring atom index, $z_i$ is the atomic number of atom $i$, and $k = 1, 2, ..., n_b$ denotes the index of the basis for each center atom type $z_i$. For a predefined distance threshold $\tau$, $\varphi(d) = 1$ if $d < \tau$ and 0 otherwise. With a reasonable $n_b$, we can enhance expressibility of the model, generating more accurate potentials. $\beta_{z_i,k}$ and $\mu_{z_i,k}$ are the learnable parameters for each atom type $z_i$, which control the center and width of each individual basis. Finally, a cosine envelope function (Thölke & De Fabritiis, 2022) $\varphi(d_{i,j})$ is applied to guarantee continuity at the cutoff edges, *i.e.*, $\frac{\partial \psi(d)}{\partial d}|_{d=\tau} = 0$:

$$\varphi(d_{i,j}) = \begin{cases} \frac{1}{2}(\cos(\frac{\pi d_{i,j}}{\tau}) + 1) & \text{if } 0 \le d_{i,j} \le \tau \\ 0 & \text{otherwise} \end{cases} \tag{3}$$

**Neighbor Embedding.** We adopt the idea of neighbor embedding (Thölke & De Fabritiis, 2022), which represents relative information from nearby atoms under the distance of some threshold $\tau$, denoted by $\mathbf{x}^{(\text{neighbor})} \in \mathbb{R}^d$:

$$\mathbf{x}_i^{(\text{neighbor})} = \sum_{j=1}^{n_b} \mathbf{U} \left[ \mathbf{x}_j^{(\text{self})} \odot \mathbf{V} \boldsymbol{\psi}_{i,j}^0 \right], \tag{4}$$

where $\boldsymbol{\psi}_{i,j} = [\psi_{i,j,1}, ..., \psi_{i,j,n_b}] \in \mathbb{R}^{n_b}$, $\mathbf{V} \in \mathbb{R}^{d \times n_b}$ is a projection matrix from radial basis functions to the atomic embedding space, and $\odot$ indicates element-wise multiplication. $\mathbf{U} \in \mathbb{R}^{d \times d}$ is another linear projection matrix. As a result, $\mathbf{x}_i^{(\text{neighbor})} \in \mathbb{R}^d$, the neighbor embedding of atom $i$, is in the same atomic embedding space. For each atom $i$, we combine the atomic and neighbor embeddings, then they are projected back to the same dimensionality by $\mathbf{W} \in \mathbb{R}^{d \times 2d}$. That is, $\mathbf{x}_i = \mathbf{W}[\mathbf{x}_i^{(\text{self})}; \mathbf{x}_i^{(\text{neighbor})}]$.

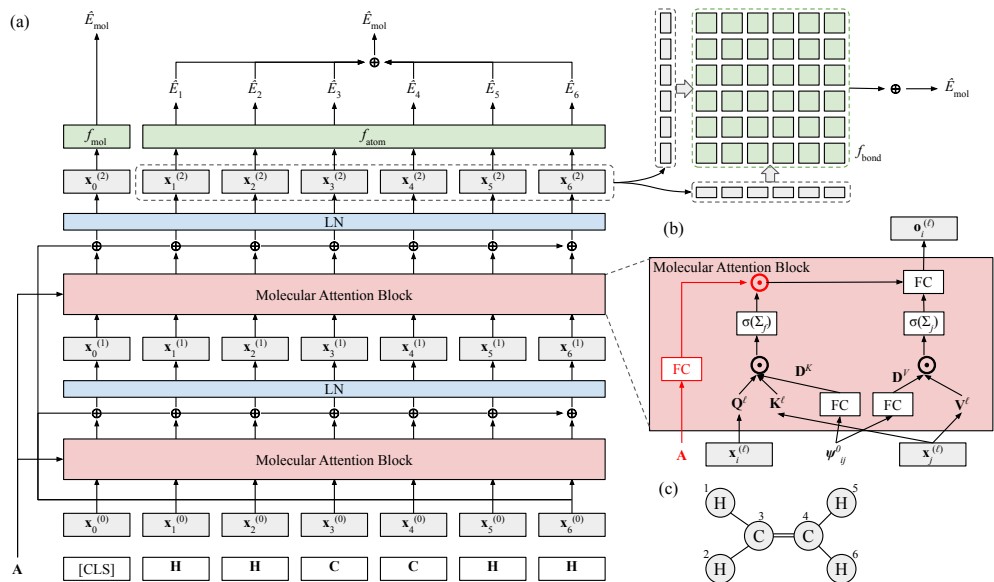

Figure 1: (a) Our model architecture. (b) Detailed Molecular Attention Block. (c) $C_2H_4$ example.

## 3.3 OUR TRANSFORMER MODEL

As illustrated in Fig. 1, our model is based on a Transformer. Given a molecule as a set of its $N$ atoms, encoded as $\mathbf{x}_i \in \mathbb{R}^d$ for $i = 1, ..., N$, our model adds an additional [CLS] token, denoted by $\mathbf{x}_0 \in \mathbb{R}^d$, to explicitly learn to represent the overall molecule embedding. On this input sequence, the model stacks $L$ Molecular Attention Blocks (MAB) to contextualize each atom representation across the molecule (within the cutoff distance $\tau$), which will be detailed subsequently. We denote by $\mathbf{x}_i^{(\ell)}$ the atom embedding after $\ell = 0, ..., L$ stages of the MABs. After $L$ blocks, the final sequence of atomic embeddings $\mathbf{x}_i^{(L)}$ are produced, and we estimate the overall molecule-level energy from them in two popular ways with Transformers. First, we may predict the atom-level energy $E_i$ for atom $i$ by passing $\mathbf{x}_i^{(L)}$ to a simple linear layer. That is, $\hat{E}_i = f_{\text{atom}}(\mathbf{x}_i^{(L)})$, where $f_{\text{atom}} : \mathbb{R}^d \to \mathbb{R}$ is an atom-level energy regressor, and then, summation over all atoms $i = 1, ..., L$ gives the molecule-level energy; that is, $\hat{E}_{\text{mol}} = \sum_{i=1}^{N} \hat{E}_i$.

Another approach is directly computing the molecule-level energy from the [CLS] by $\hat{E}_{\text{mol}} = f_{\text{mol}}(\mathbf{x}_0^{(L)})$, where $f_{\text{mol}} : \mathbb{R}^d \to \mathbb{R}$ is a molecule-level energy regressor. For $f_{\text{atom}}$ and $f_{\text{mol}}$, we use a single fully-connected layer. Both approaches are evaluated in Sec. 4. In Sec. 3.4, we will introduce our main approach for this regression to take advantage of domain knowledge from physics.

**Details on Molecular Attention Block.** Each Molecular Attention Block (MAB) at level $\ell$ takes a sequence of atomic embeddings $\{\mathbf{x}_i^{(\ell-1)} : i = 0, ..., N\}$ from the previous level. For each atom $\mathbf{x}_i^{(\ell-1)}$ as query and all atoms including $i$ as the context (keys and values), it performs self-attention as in Fig. 1(b). Following TorchMDNet (Thölke & De Fabritiis, 2022), we modify from the vanilla Transformer (Vaswani et al., 2017) to explicitly reflect the relation arisen from the physical distance between two atoms $i$ and $j$, in addition to the semantic relevance between them modeled by regular Transformers. Specifically, from the radial basis $\psi_{i,j}^0$ (Orr et al., 1996), we compute $\mathbf{D}^K, \mathbf{D}^V \in \mathbb{R}^{N \times N \times m}$, where $m$ is the embedding dimensionality used for query, key, and value. An element $d_{i,j}^K, d_{i,j}^V \in \mathbb{R}^m$ represents physical tendency to attract each other between atom $i$ and $j$ for key-purpose and value-purpose, respectively. These are mapped from the radial basis function $\psi_{i,j}^0$ by a linear layer, followed by SiLU (Elfwing et al., 2018) activation. This relation is represented as $\mathbb{R}^m$ instead of a scalar to reflect dimension-wise relationship.

In addition to the changes introduced by Thölke & De Fabritiis (2022), we additionally feed the adjacency matrix $\mathbf{A} \in \{0, 1\}^{N \times N}$, followed by a linear layer and SiLU activation. This $\mathbf{A}$-mask is multiplied element-wise with the inferred attention weights, in order to additionally control this semantic relevance based on physical adjacency. For instance, two atoms that are far away will be likely multiplied by a low value, reducing its relationship even if semantic relevance is estimated high. This part can be optionally turned on or off, and we provide ablation study in Sec. 4.

### 3.4 Physics-driven Parametric Energy Prediction

Instead of directly regressing to the atom or molecule energy as described in Sec. 3.3, we propose to design a parametric model that reflects physical insights. For this formulation, we use a simple form that can simultaneously reflect the repulsive and attractive forces between two atoms $i, j$ within the bond energy $E_{i,j}$; namely, Coulomb's law and Lennard-Jones Potential (LJP):

$$E_{i,j} = -\beta_1 \frac{\beta_0}{d_{i,j}} + \beta_2 \left[ \left( \frac{\beta_4}{d_{i,j}} \right)^{2\beta_3} - 2 \left( \frac{\beta_4}{d_{i,j}} \right)^{\beta_3} \right], \tag{5}$$

where $\beta_0$ corresponds to $q_i q_j$, influence of charges between two atoms in Coulomb potential. $\beta_4$ is the equilibrium distance between atom $i$ and $j$, where the repulsive and attractive forces become equivalent, and thus the atom-atom potential energy becomes zero. The energy becomes minimal at this point. $\beta_1$ and $\beta_2$ are linear coefficients for the Coulomb and LJP parts. It is known that $\beta_3 \approx 6$ under the condition of following the London dispersion force London (1930); Cornell et al. (1995), but the repulsive equivalence of $2\beta_3 \approx 12$ is much more an approximate term (square of the attractive term), so we leave $\beta_3$ as an open parameter to be learned from the given data. These 5 parameters, denoted by $\boldsymbol{\beta} = [\beta_0, \beta_1, \beta_2, \beta_3, \beta_4]$, are estimated by a regressor $f_{\text{bond}} : \mathbb{R}^{2d} \to \mathbb{R}^5$; that is, $\hat{\boldsymbol{\beta}} = f_{\text{bond}}([\mathbf{x}_i; \mathbf{x}_j])$.

The overall molecule-level energy is calculated by the sum of all pair-wise bond energies and the atomic self-energies; that is, $\hat{E}_{\text{mol}} = \hat{E}_{\text{bond}} + \hat{E}_{\text{atom}}$, where $\hat{E}_{\text{bond}}$ and $\hat{E}_{\text{atom}}$ are defined as

$$\hat{E}_{\text{bond}} = \sum_{i=1}^{N} \sum_{j>i}^{N} \hat{E}_{i,j}, \quad \text{and} \quad \hat{E}_{\text{atom}} = \sum_{i=1}^{N} f_{\text{atom}} \left( \mathbf{x}_i^{(L)} \right). \tag{6}$$

We summarize what to expect from modeling to predict the parameters of a physical formulation as follows. First, we aim to make sure that we satisfy physical conditions when we learn from given information and extrapolate to unseen cases. Second, by observing the predicted parameters, we can monitor whether the model actually captures the physical properties of the molecule. Lastly, we expect the model to predict the energy directly from $\hat{E}_{\text{atom}}$ if the given formula is difficult to follow. In Eq. (5), for instance, if the inter-atomic potential does not fit well with LJP, the model assigns $\beta_3 \approx 0$, relying solely on the Coulombic potential.

Initially, the model is subjected to minimize the MSE Loss between the predicted molecule energy $\hat{E}_{\text{mol}}$ and its ground truth $E_{\text{mol}}$. In other words, the energy loss $\mathcal{L}_{\text{energy}}$ is defined as $\|\hat{E}_{\text{mol}} - E_{\text{mol}}\|^2$.

### 3.5 Masked Atomic Modeling

Masked Language Modeling (MLM), originally introduced by BERT (Devlin et al., 2018) for language modeling, has been successfully utilized as a pre-training task for various multimodal models and tasks (Lu et al., 2019; Sun et al., 2019; Zhang et al., 2020a). The main idea is to randomly mask a subset of tokens and let the model recover them from its contexts, *i.e.*, the other textual or visual tokens in the input sequence. This concept naturally supports self-supervised learning as long as the elements in the input sequence are contextually relevant, requiring no human labeling.

In this paper, we propose Masked Atomic Modeling (MAM) in a similar spirit. All chemical materials are composed of multiple atoms, often with more than one type. When a majority of atoms in a valid molecule is known, a set of possible atoms in the rest is significantly reduced when considering the properties of each atom according to the law of chemistry, *e.g.*, the octet rule, Lewis symbol analysis. With MAM, we would like to train our Transformer to discover such chemical restrictions purely by observing a set of valid molecules in the training examples without direct supervision.

Formally, on a sequence $\mathbf{X} \in \mathbb{R}^{N \times d}$ with $N$ atoms, we randomly mask each token by a probability of $\rho$ (we use 0.3, twice as Devlin et al. (2018)), replacing the masked tokens to `[mask]`. In the resulting matrix, the rows of $\mathbf{X}$ corresponding to the masked tokens, are replaced with the mask embedding. The model is trained to minimize the log loss over the masked tokens:

$$\mathcal{L}_{\text{mask}} = -\log p(\mathbf{X} \otimes \mathbf{m} | \mathbf{X} \otimes (\mathbf{1} - \mathbf{m})) \tag{7}$$

where $\mathbf{m} \in \{0, 1\}^N$ is a binary mask vector for atoms, $\mathbf{1}$ is a one-valued vector, and $\otimes$ indicates row-wise multiplication. $p$ is estimated by a binary classifier, where we use a two-layer MLP.

### 3.6 COMBINING PHYSICAL CONSTRAINTS

**Zero-Force Regularization.** When a molecule is in its equilibrium state, the net force on each atom should be at zero. This condition may provide a strong hint for the model to find the valid and optimal molecular structure, but this has not been utilized well in existing studies. Thus, we additionally regularize to minimize the force, computed by the partial gradients of the predicted energy with respect to the 3-dimensional axis $(x, y, z)$. Formally,

$$\mathcal{L}_{\text{force}} = \sum_{i=1}^{N} \|\hat{\mathbf{F}}_i\|^2 = \sum_{i=1}^{N} \left( \frac{\partial E_i}{\partial x} \right)^2 + \left( \frac{\partial E_i}{\partial y} \right)^2 + \left( \frac{\partial E_i}{\partial z} \right)^2, \tag{8}$$

where $\hat{\mathbf{F}} \in \mathbb{R}^3$ is the predicted force of atom $i$.

**Inequality Bound Condition.** The definition of a stable equilibrium structure is a structure corresponding to the lowest energy under the given composition. Such an optimal structure can be found by estimating energy from the given structure, differentiating it with respect to the position, and changing the position based on force information. Naturally, if there is any local deviation from the optimal structure, the energy is always greater than its ground-state energy. This sounds obvious physically, but a machine learning model is unaware of this and its estimation may be invalid. Therefore, we apply an additional condition to narrow down the solution space. The energy should be greater than the ground state when locally deviating from the stable structure. During training, small Gaussian random noise with an amplitude of 0.5 Å is applied to the optimal structure. This is implemented by an additional loss term $\mathcal{L}_{\text{bound}}$ based on the energy inequality condition:

$$\mathcal{L}_{\text{bound}} = \begin{cases} \hat{E}_{\text{mol}} - \hat{E}_{\text{mol}}^* & \text{if } \hat{E}_{\text{mol}}^* \leq \hat{E}_{\text{mol}} \\ 0 & \text{otherwise} \end{cases} \tag{9}$$

### 3.7 OVERALL OBJECTIVE

Combining all together, our model minimizes three loss functions:

$$\mathcal{L} = \mathcal{L}_{\text{energy}} + \lambda_{\text{mask}} \mathcal{L}_{\text{mask}} + \lambda_{\text{force}} \mathcal{L}_{\text{force}} + \lambda_{\text{bound}} \mathcal{L}_{\text{bound}}, \tag{10}$$

where $\lambda_{\text{force}}$, $\lambda_{\text{mask}}$, and $\lambda_{\text{bound}}$ are coefficients controlling relative importance of each loss term.

## 4 EXPERIMENTS AND RESULTS

We conduct experiments to answer the following questions: **Q1**. How does our model perform on energy estimation compared to other models? (Sec. 4.2) **Q2**. How does our model perform on a basic structural optimization experiment compared to other models? (Sec. 4.3) **Q3**. How much physics-driven constraints affect the prediction? (Sec. C)

### 4.1 EXPERIMENTAL SETTINGS

**Datasets.** We use two public benchmarks to compare our model with competing models. QM9 (Ruddigkeit et al., 2012; Ramakrishnan et al., 2014) dataset is a collection of optimal structures of 130,000 molecules with up to nine atoms of {C, H, O, N, F}, selected from GDB-17 (Ruddigkeit et al., 2012). We use 80% for training, 5% for validation, and 15% for testing. Additionally, we evaluate on OC20 dataset (Chanussot* et al., 2021), which contains stable structures and relaxation trajectories for systems of 15K bulk catalysts and 82 adsorbates. We train our model on two tasks, energy and force regression based on the given structure (S2EF) and relaxed energy prediction with given initial structure (IS2RE).

**Baselines.** We compare state-of-the-art energy prediction models on the QM9 dataset: SchNet (Schütt et al., 2018), DimeNet (Gasteiger et al., 2020), TorchMDNet(ET) (Thölke & De Fabritiis, 2022), ForceNet (Hu et al., 2021) and MXMNet (Zhang et al., 2020b). Except for ForceNet and MXMNet, publicly available pre-trained parameters are utilized in the experiment.

**Evaluation Metric.** We report the mean average error (MAE) between the prediction and the ground truth for the estimation of energy ($\text{MAE}_{\text{E}}$, in meV/mol) and force ($\text{MAE}_{\text{F}}$, in eV/Å) estimation, following existing studies.

More implementation details are provided in Sec. A in the Appendix.

| Dataset (Task) | QM9 | | | OC20 (S2EF) | OC20 (IS2RE) | |
| Model | $MAE_E$ ($\downarrow$) | $MAE_F$ ($\downarrow$) | $\Delta P$ (Å) ($\downarrow$) | $MAE_F$ ($\downarrow$) | $MAE_F$ ($\downarrow$) | $\Delta P$ (Å) ($\downarrow$) |
|---|---|---|---|---|---|---|
| SchNet Schütt et al. (2018) | 14.00 | 2.64 | 0.47 | 0.0743 | 1.059 | 0.60 |
| CGCNN Xie & Grossman (2018) | – | – | – | 0.0673 | 0.988 | 0.58 |
| MXMNet Zhang et al. (2020b) | **5.90** | 1.83 | 1.57 | – | – | – |
| DimeNet Gasteiger et al. (2020) | 8.02 | 1.79 | 0.58 | 0.0693 | 1.012 | 0.55 |
| ForceNet Hu et al. (2021) | 18.62 | 0.41 | 0.21 | – | – | – |
| TorchMDNet (ET) Thölke & De Fabritiis (2022) | 6.15 | 1.15 | 0.32 | – | – | – |
| GemNet-dT Gasteiger et al. (2021) | – | – | – | 0.0257* | – | 0.18 |
| SpinConv Shuaibi et al. (2021) | 12.00 | – | – | 0.0329* | – | 0.21 |
| Ours ($\mathcal{L}_{energy}$ only) | 8.35 | 1.28 | 1.23 | – | – | – |
| Ours (full model) | 15.16 | **0.005** | **0.025** | **0.0549** | **0.887** | **0.10** |

Table 1: Comparison with baseline models for energy and force accuracy (in MAE) and average distortion $\Delta P$ after structure optimization experiment. (*indicates trained on 100× larger data.)

## 4.2 COMPARISON WITH BASELINES

Tab. 1 compares the performance of our model with baselines on tasks of energy prediction, force prediction, and structure optimization. We report the performance of our model with two configurations; one with the full model and the other with $\mathcal{L}_{energy}$ only, by setting $\lambda_{\{mask,force,bound\}}$ to zero.

In this line of research, the Energy MAE has been the most widely used metric. At a glance to the first column, Energy MAE, we observe that our proposed model estimates the molecule energy comparably with most baselines, slightly lagging behind the current state-of-the-art, MXMNet.

We ask a question here: does this mean that the models actually understand the molecular structure and estimate the energy precisely from it? We are not able to answer this question solely from the energy MAE, since one energy value in the optimal structure does not contain all information on how the PES is reflected according to the degree of freedom with the position of the atoms. Therefore, we further investigate with additional metrics which reflect the structure or physical conditions: zero-force condition on optimal structures and simple structural optimization task.

The second column of Tab. 1 reports the MAE in force estimation of each model by differentiating the energy with respect to the position. Ideally, lower energy MAE should indicate better molecular structure, and thus it should lead to lower force MAE as well. This is the underlying expectation that the entire research community has solely focused on energy MAE optimization.

Interestingly, we observe that energy MAE and force MAE do not have strong one-to-one correspondence. All existing models and our $\mathcal{L}_{energy}$ model is optimized only for the energy prediction while showing significantly worse performance on the force estimation. In other words, those previous models are significantly over-optimized only for energy estimation, without consideration of basic constraints that potential models must satisfy. On the other hand, our full model dramatically reduces the force error through the additional terms based on physical insights. Although the energy MAE is slightly higher than other models like MXMNet, a difference of about 10 meV/mol is acceptable in terms of the overall quality.

On OC20, We experiment with both tasks, S2EF and IS2RE. The result is added to Tab. 1, where we compare against a few baselines using scores reported in Open-Catalyst-Project[1]. This table indicates that our method is competent on both tasks, outperforming all baselines.

## 4.3 QUALITATIVE ANALYSIS WITH STRUCTURE OPTIMIZATION

A primary application for calculating molecular energy is to search for a stable structure and to perform molecular dynamics (MD) simulations of structural changes over temperature and time. All of these works are the foundation for the design and discovery of new materials (Friederich et al., 2021; Louie et al., 2021).

In order to see if the models actually capture the optimal structure of molecules, we conduct an additional structure optimization experiment. Starting from the stable structures in QM9, we slightly perturb each atom's position from its original optimum and optimize the structure again, expecting it to converge back to the original optimal one. Upon convergence, we measure the structure distortions $\Delta P$, in average Euclidean distance, as each atom moves from its optimal structure.

---

[1] https://github.com/Open-Catalyst-Project

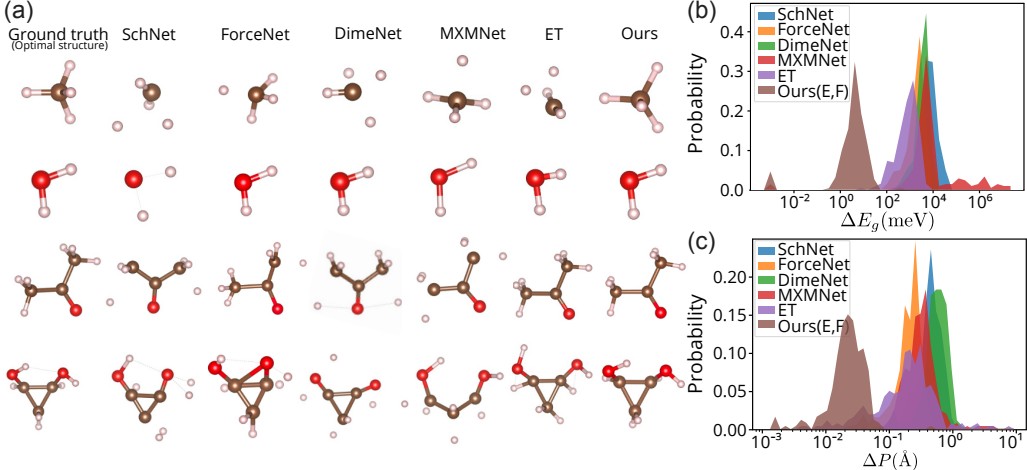

Figure 2: (a) Structural optimization results. The left-most column is the initial stable structure in QM9, followed by recovery results by competing models sequentially. For more structural optimization results, see Appendix Fig. I. (b-c) Distribution of energy difference ($\Delta E_g$) and structural change ($\Delta P$) before and after structural optimization, in log scale.

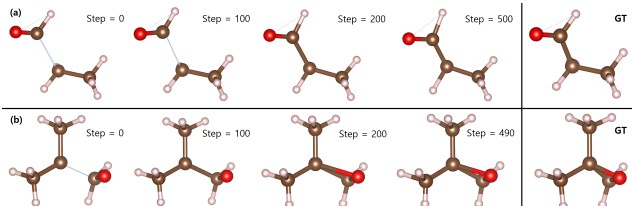

Figure 3: Molecule assembly results from (a) GDB-35 and (b) GDB-87. GT is the ground truth structure with 500 relaxation steps. The disconnected bond is well connected and recovered to its original stable structure.

The right-most column of Tab. 1 reports quantitative comparison results of this experiment. Our physics-driven model achieves a more accurate score compared to other models, indicating that our proposed model indeed understands and is capable of reproducing a stable structure instead of simply over-optimizing on the energy estimation.

Fig. 2 (a) illustrates a few examples of optimized structures by baseline models and ours. The initial stable structures in the left-most column are not well-maintained by the baselines. For the simplest molecule, $CH_4$, the Hydrogen atoms around the Carbon atom are expected to be symmetrically arranged, but the optimized structures by the baselines are not symmetric. In more complex molecules, most potentials do not preserve the stable structure.

Fig. 2 (b,c) show the distribution of the energy difference $\Delta E_g$ before and after the reoptimization, averaged over 256 molecules (128 smallest and 128 randomly sampled molecules) from the QM9 dataset. Fig. 2 (b) demonstrates the $\Delta E_g$ in log scale, which clearly shows the difference in the performance of each model. With our model, the center of $\Delta E_g$ is at least two orders of magnitude smaller compared to other potentials, indicating that our model recovers the optimal structure far better than all other models. Also, in Fig. 2 (c), the distance deviation $\Delta P$ is mostly less than 0.1Å, and even compared to other potentials, it can be seen that $\Delta P$ is at least 10 times smaller. This structure optimization benchmark is a tough task because the QM9 dataset consists only of stable structures. Furthermore, it is unreasonable to expect molecular dynamics (MD) to work appropriately, reflecting the temperature and dynamics on this stable structure-only dataset. On the contrary, great performance on this challenging optimization task demonstrates that our physics-driven model captures basic physical information, such as distance symmetry, even from limited information. Additional examples are presented in Sec. D in the Appendix.

### 4.4 MOLECULE ASSEMBLY TASK

We employ our approach for an additional task, namely, the molecule assembly task. In addition to the structure optimization conducted in Sec. 4.3, where we start from (almost) optimal structure and optimize the energy to see if the model can recover the stable structure again, this molecule assembly task makes it further challenging by even breaking one or more bonds in the molecule by moving some functional group far away. This task aims to recover the original stable structure from this

| No. | Base | [CLS] | LJP | Mask | Force | Bound | MAE$_E$ ↓ | MAE$_F$ ↓ | $\Delta P$ ↓ |
|---|---|---|---|---|---|---|---|---|---|
| 1 | ✓ | | | | | | 11.83 | 0.77 | 1.76 |
| 2 | ✓ | ✓ | | | | | 9.03 | 0.90 | 1.11 |
| 3 | ✓ | ✓ | ✓ | ✓ | | | 9.70 | 1.91 | 0.814 |
| 4 | ✓ | ✓ | ✓ | | ✓ | | 10.18 | 0.016 | 0.141 |
| 5 | ✓ | ✓ | ✓ | | | ✓ | 16.34 | 0.007 | 0.038 |
| 6 | ✓ | ✓ | | ✓ | ✓ | ✓ | 20.67 | 0.004 | 0.022 |
| 7 | ✓ | ✓ | ✓ | | ✓ | ✓ | 17.50 | 0.005 | 0.027 |
| 8 | ✓ | ✓ | ✓ | ✓ | | ✓ | 17.34 | 0.013 | 0.044 |
| 9 | ✓ | ✓ | ✓ | ✓ | ✓ | | 9.65 | 0.015 | 0.083 |
| 10 | ✓ | ✓ | ✓ | ✓ | ✓ | ✓ | 15.16 | 0.005 | 0.025 |

Table 2: Ablation study results, adding or subtracting components in the loss function. Red indicates unacceptably inferior results (MAE$_F$, $\Delta P \gg 0.1$).

completely broken one. Since the QM9 dataset does not contain non-equilibrium information, it is challenging to expect accurate energy values along the pathways in which molecules are combined. Therefore, our goal is to examine if the bond-broken molecule can be re-assembled to a stable structure. For this, we randomly select one or two functional groups in a molecule and disconnect bonds between them by translating each towards different directions. (In our experiment, we move them by 0.7Å.) Starting from this distorted structure, we optimize the structure again using our model to see if it can recover the original stable structure.

As shown in Fig. 3, only our method succeeds in recovering the structure to the original undistorted one, while others show catastrophic failure. To this end, we conclude that the strength of our approach to reproduce a physically correct molecular structure can be extended to reproducing the correct reaction pathway.

### 4.5 ABLATION STUDY

We conduct ablation study to see which component contributes improving which metric. The 'Base' model indicates our Transformer model described in Sec. 3.3 without using [CLS] token. Tab. 2 compares multiple configurations of our model using a subset of components. Comparing #1 and #2, the [CLS] token turns out to be effective, reducing the energy error. The rest compares by adding each component separately starting from our base+equation model (#3, #4, #5) and by eliminating each component from the full model (#6 - #10). We observe the following:

- **Mask** plays its role in improving the energy estimation. Comparing #7 and #10, having Mask helps the model to improve MAE$_E$ without affecting MAE$_F$ or $\Delta P$. Solely with Mask (#3), it achieves a nice MAE$_E$, but its structure looks suboptimal implied by inferior MAE$_F$ and $\Delta P$.
- **Bound** condition is the most important component for understanding the overall structure. Without it (#9), $\Delta P$ gets significantly worse than the full model (#10), while MAE$_E$ gets (probably illegally) better by focusing more on the energy like baseline models. With Bound only (#5), it achieves reasonable MAE$_F$ and $\Delta P$, which is not possible only with Mask (#3) or Force (#4).
- **Force** affects all three metrics slightly at the same time. Without Force (#8), all three metrics get slightly worse compared to the full model (#10). With the Force only (#4), however, the $\Delta P$ is suboptimal. We can conclude that the Bound condition is also needed to get the acceptable $\Delta P$.

Sec. B in the Appendix presents additional ablation study on model size and MAM masking ratio. Also, Sec. C provides additional qualitative analysis on MAM and physics-driven modeling.

## 5 CONCLUSION

In this work, we propose a physics-driven model and regularization scheme for molecular energy prediction. The physics-driven design enables our model to generate a physically meaningful structure. We also propose a self-supervised learning method inspired by masked language modeling. Our observation indicates that the state-of-the-art models may not be robust under the structure optimization task with the QM9 dataset, which only contains optimized stable structures. We utilize inequality constraints and force information of the optimal conformations, which results in physically more reasonable outcomes. With the combination of physics-driven modeling and regularization, our model outperforms the state-of-the-art models in structure optimization tasks with the QM9 dataset. We also demonstrate that our approach can be used for structure optimization with a few meV scale differences in the initial structure of small molecules. Furthermore, we find that much improvements are needed to achieve a reasonable PES that not only matches a single energy value for large molecules. When constructing an ML potential, it is always challenging to have a sufficient dataset; thus, our approach that maximally utilizes the information in both model design and training would shed light on future research.

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
