# OpenReview forum: "Physics-empowered Molecular Representation Learning"
_ICLR.cc/2023/Conference — Submitted to ICLR 2023_

### Official Review · Reviewer_x1A4 · 2022-10-19

**Confidence:** 4
**Correctness:** 3
**Technical Novelty And Significance:** 2
**Empirical Novelty And Significance:** 2
**Recommendation:** 3

**Clarity, Quality, Novelty And Reproducibility:**

The manuscript is well written, easy to understand, and the results are clear. Part of the approach cannot be considered novel, as  outlined above.

**Strength And Weaknesses:**

Strength:
1) the idea of training  a network using a BERT-style censoring of the position of some atoms is, to my knowledge, original.

Weaknesses:
1) the challenge in the field is reproducing the forces (and the energies) along a *reaction pathway*, namely for sequences of configurations connecting two different molecular species, or two different configurations of the same molecule. The forces along these pathways are typically very high, and determine the chemistry of the system.  The test set used in this work is therefore mostly of academic interest: the forcefield developed in this work is good for reproducing the geometry of the molecules, but worse than other approaches in reproducing the energy difference between molecules. No relevant "activated" geometry is included in the test set.
2) the idea of fitting the forces, and not only the energy, dates back to the 90ties (see wiki, "force matching") and is  used almost ubiquitously also in NN training (see, for example, https://pubs.acs.org/doi/10.1021/acscentsci.8b00913 eq 9). The fact that by adding a penalty on the value of the forces allows recovering the correct  geometry is correct, but trivial, since the configurations explored in these optimization are very similar to the global minimum. The relevant result would be reproducing the forces along a reaction pathway (see point 1).
3) The idea of adding explicitly LJ or Coulomb interactions in a NN is also not new, but, more importantly, adding these terms for isolated and very small molecules is not chemically justified. In fact these terms are typically added to describe liquid environments (see for example https://www.nature.com/articles/s41467-020-20427-2 ), where different molecules interact by non-bonded interactions.

**Summary Of The Paper:**

The manuscript describes a Transformer-based neural network aimed at predicting the energies and the forces of molecules. The network is trained on the prediction of the energy of ~130,000 small molecules. The loss includes the square deviation between the ground truth energy and the predicted energy, and also terms aimed at ensuring that the "experimental" structures are stable.  The energy estimator includes, beside the transformer, a  two-body energy, eq 5. The model predicts the energy with an accuracy which is marginally worse than that of other approaches, but it allows recovering the correct structure in geometry optimizations started from perturbed configurations.

**Summary Of The Review:**

The results presented in the manuscript are mostly technically correct, but, in my opinion,  do not advance significantly the state of the art of machine-learning based potentials, since the numerical tests are performed on tasks which are not of practical use in molecular modelling. Some of the directions which are explored , for example the BERT-style masking loss, are interesting and could bring to important advances.

---

> ### Author Response · Authors · 2022-11-19
> **Answers to Reviewer x1A4 (3)**
>
> **Q2: The idea of fitting the forces is not new.**
>
> A2: Yes, it is by no means the first time that we have used force itself for NN training, and we are well aware of it. However, we would like to point out that we apply new physical constraints defining optimal structure which has not been tried before: under the optimal structure, all atomic forces are zeros and total energy of a molecule is lower than locally distorted structure. These are simple but solid physical constraints, and we verify that these indeed lead to better empirical results, especially to optimize structure beyond the training distribution. We utilize these conditions for the first time to the best of our knowledge, and we believe it will be valuable information for readers when using the numerous optimal structure data mentioned above.
>
> Considering your comments, we employ our approach for the molecular assembly task (see our answer above for Q1-1), and only our method succeeds in recovering the stable structure from a largely undistorted one with broken bonds, while others show catastrophic failure. In this sense, we believe the physical conditions we introduce are simple but still novel and effective.
>
>
> **Q3: Adding LJ or Coulomb interactions for isolated and very small molecules is not chemically justified.**
>
> A3: As the reviewer mentioned, many existing efforts have tried extra interaction terms considering short-range and long-range interactions. The purpose of LJ or Coulomb terms in this line of work is to represent the accurate physical behavior of molecular interactions with proper numerical terms. In our work, in addition to this, we introduce them to extrapolate the model prediction beyond the training distribution. By predicting equation parameters from a neural-network rather than predicting energy directly, our model extrapolates energy prediction beyond the range of the training set, unlike other typical neural-net models that easily overfit and generalize only within the training distribution. For the sake of generalizability, we choose to fit energy to simpler LJ and Coulomb equations which are reasonable terms as a first step towards regularizing the behavior of interatomic interaction in a physics-informed way. It is true that coulomb potential has been added to describe liquid environments in the referenced paper, but this work is essentially based on an equation-free method where the short-range interaction is described by the equation-free method, and the extra coulomb interaction term is added to describe long-range interaction. Therefore, the purpose of the Coulomb term represented in the referenced paper is fundamentally different from ours, where whole energy is represented as LJ and Coulomb terms mainly for extrapolating energy prediction of out-of-distribution training data.
>
> We are more than happy to answer your additional questions and open for further discussion. Please feel free to reply. Thanks again for your constructive feedback.

---

> ### Author Response · Authors · 2022-11-19
> **Answers to Reviewer x1A4 (2)**
>
> **Q1-2: The proposed method is better at geometry reproduction but not on energy prediction.**
>
> A1-2: Yes, at a glance, it looks that our proposed model is stronger in force prediction but slightly weaker in energy estimation. However, our experiment reveals that previous works only predicted the energy accurately but are not able to generate a legitimate potential energy surface (PES). This is a result of *over-optimization without proper constraints on underlying physics*, e.g., valid molecular structure. This is supported by their significantly higher MAE_F ($\text{MAE}_\text{F}$) and measured structure distortion ($\Delta P$) in Tab. 1, and broken molecular structures they converged, illustrated in Fig. 2 and III in Appendix. These are not physically feasible and incorrect. The energy based on this invalid PES is just simulated to be slightly closer to the GT, but their solution is not within valid physical conditions. In contrast, our model preserves the actual structure, indicating the energy difference we achieve is the actual, valid state-of-the-art.
>
> Tab. 1 is copied below for your convenience:
>
> | Dataset (Task)  |  QM9  |  |  |  OC20 (S2EF)  |  |  OC20 (IS2RE) |
> |-----------------------------------------|:-------------------------------:|:-------------------------------:|:-------------------------:|:-------------------------------:|:-------------------------------:|:-------------------------:|
> | Model  | MAE$_\mathrm{E}$ ($\downarrow$) | MAE$_\mathrm{F}$ ($\downarrow$) | $\Delta P$ ($\downarrow$) | MAE$_\mathrm{F}$ ($\downarrow$) | MAE$_\mathrm{F}$ ($\downarrow$) | $\Delta P$ ($\downarrow$) |
> | SchNet   |  14.00  |  2.64  |  0.47  |  0.0743  |  1.059  |  0.60  |
> | CGCNN  |  -  |  -  |  -  |  0.0673  |  0.988  |  0.58  |
> | MXMNet   |  5.90  |  1.83  |  1.57  |  -  |  -  |  -  |
> | DimeNet  |  8.02  |  1.79  |  0.58  |  0.0693  |  1.012  |  0.55  |
> | ForceNet  |  18.62  |  0.41  |  0.21  |  -  |  -  |  -  |
> | TorchMDNet (ET)  |  6.15  |  1.15  |  0.32  |  -  |  -  |  -  |
> | GemNet-dT  |  -  |  -  |  -  |  0.0257*  |  -  |  0.18  |
> | SpinConv  |  12.00  |  -  |  -  |  0.0329*  |  -  |  0.21  |
> | Ours ($\mathcal{L}_\text{energy}$ only) |  8.35  |  1.28  |  1.23  |  -  |  -  |  -  |
> | Ours (full model)  |  15.16  |  0.005  |  0.025  |  0.0549  |  0.887  |  0.10  |

---

> ### Author Response · Authors · 2022-11-19
> **Answers to Reviewer x1A4 (1)**
>
> We appreciate the reviewer for insightful comments and feedback. We answer to each question below:
>
> **Q1-1: Reproducing the whole reaction pathway is important.**
>
> A1-1: It is certainly a feasible and important research direction to reproduce the energy and force along the reaction pathway by including the reaction scenario in the training set. We will work along this line as a continuation of this work. However, we would like to emphasize that atomic-level simulations are conducted for various purposes such as finding optimal structures, molecular dynamics, kinetic Monte-Carlo, searching meta-stable structures, inverse material design, and more. In these cases, it does not necessarily include abrupt chemical reactions. Mimicking chemical reactions is challenging, and there are various attempts to consider the trade-off between speed and accuracy depending on the scope and interest. We believe that the expectation and utilization of neural networks can be broader; for instance, even with the classical force field alone, early versions and recent ReaxFF that handle chemical reactions arise, but ReaxFF does not replace all classical force fields.
>
> Especially with advances in machine learning, we are slowly getting closer to inverse material design, which is considered as a dream [A]. Establishing massive material datasets is a foundational step for material design.
> A prime example of this would be the Material-project, Aflow, which contains millions of stable structures. It is hard to deny that utilizing the vast amount of optimal structure data should be encouraged, even if there are no strict chemical reaction pathway structures.
>
> Although reproducing the whole reaction pathway is challenging within the scope of our work, by considering the reviewer’s constructive feedback, we conduct an additional feasibility study along this line by employing our approach for the molecular assembly task. This is to optimize the molecular structure given a distorted molecular geometry with a few disconnected bonds. For this, we randomly select one or two functional groups in a molecule and disconnect bonds between them by translating each towards different directions. (In our experiment, we move them by 0.7A.) Starting from this distorted structure, we optimize the structure again using our model to see if it can recover the original stable structure. As shown in Fig. 3 (in Sec 4.4), our method successfully recovers the structure to the original one. Other models, on the other hand, have already suffered from a catastrophic failure, even starting from an almost optimal structure. To this end, the strength of our approach to reproduce physically correct molecular geometry can be potentially extended to reproduce the correct reaction pathway by applying a large scale dataset containing activated geometries. This is a promising and interesting future direction.
>
> [A] Sanchez-Lengeling, Benjamin, and Alan Aspuru-Guzik. "Inverse molecular design using machine learning: Generative models for matter engineering." Science 361.6400 (2018): 360-365.

---

### Official Review · Reviewer_bSWh · 2022-10-23

**Confidence:** 3
**Correctness:** 3
**Technical Novelty And Significance:** 3
**Empirical Novelty And Significance:** 3
**Recommendation:** 5

**Clarity, Quality, Novelty And Reproducibility:**

The paper is well written.
There are missing details such hyperparameters.

**Strength And Weaknesses:**

Strengths:
1. This paper applies masked atomic modeling to molecular modeling, similar to masked language modeling. The most important part is that the model is not just naively using a standard transformer but incorporates a lot of problem-specific design.
2. This paper shows good empirical results compared to previous methods.

Weaknesses:
1. The dataset and the molecules in the study are relatively small. Also, all experiments are conducted on the same dataset for one task. It is questionable how such learned representations can be used for other datasets/tasks.
2. The ablation study is not clear. It only shows several extreme comparisons of removing several important loss function terms. However, it is questionable how each design component is effective. For example, what is the optimal mask ratio?

**Summary Of The Paper:**

This paper proposes a novel masked atomic modeling framework for molecular representation learning. Their architecture combines several problem-specific designs, including Molecular Attention Block and other physics constraints as loss functions, including Zero-Force Regularization and Bound Condition. Empirical results show that the proposed method outperforms previous popular frameworks such as DimeNet.

**Summary Of The Review:**

This paper proposes Masked atomic modeling for molecular representation learning. It also incorporates several physical constraints effectively. I think this is solid contribution to the community.

---

> ### Author Response · Authors · 2022-11-19
> **Answers to Reviewer bSWh (2)**
>
> **Q2-1: Improve ablation study to see how each design component is effective?**
>
> A2-1: We refresh the ablation study in Tab. 2, by adding each component separately starting from our base+equation model (#3, #4, #5), and by eliminating each component from the full model (#6 - #10). We use a 6-layered model for this ablation.
> * Mask plays its role in improving the energy estimation. Comparing #7 and #10, having Mask helps the model to improve $\text{MAE}_\text{E}$ without affecting $\text{MAE}_\text{F}$ or $\Delta P$. Looking at #3, solely with Mask, it achieves a nice $\text{MAE}_\text{E}$, but its structure looks suboptimal implied by inferior $\text{MAE}_\text{F}$ and $\Delta P$. By masking parts of the molecule, the model is forced to learn which atoms to be placed in the masked position. This seems to help the model to predict energy better, but solely with this it may not be aware of the entire structure.
> * Bound condition is the most important component for understanding the overall structure. Without it (#9), $\Delta P$ gets significantly worse compared to the full model (#10), while $\text{MAE}_\text{E}$ gets (probably illegally) better by focusing more on the energy itself like baseline models. With the bound condition only (#5), it achieves reasonable $\text{MAE}_\text{F}$ and $\Delta P$, which is not possible with using only Mask (#3) or Force(#4).
> * Force affects all three metrics slightly at the same time. Without Force (#8), all three metrics get slightly worse compared to the full model (#10). With the Force only (#4), however, the $\Delta P$ is suboptimal. We can conclude that the Bound condition is also needed to get a reasonable range of $\Delta P$.
>
> From this observation, we conclude that Mask, Force, and Bound play their own role to improve different metrics.
>
> Tab. 2 is copied below for your convenience:
>
> | No. |  Base  |  [CLS]  |  LJP  |  Mask  |  Force  |  Bound  | MAE$_\mathrm{E}$ ($\downarrow$) | MAE$_\mathrm{F}$ ($\downarrow$) | $\Delta P$  ($\downarrow$) |
> |:---:|:------------:|:------------:|:------------:|:------------:|:------------:|:------------:|:-------------------------------:|:-------------------------------:|:--------------------------------------------:|
> |  1  | $\checkmark$ |  |  |  |  |  |  11.83  |  0.77  |  1.76  |
> |  2  | $\checkmark$ | $\checkmark$ |  |  |  |  |  9.03  |  0.90  |  1.11  |
> |  3  | $\checkmark$ | $\checkmark$ | $\checkmark$ | $\checkmark$ |  |  |  9.70  |  1.91  |  0.814  |
> |  4  | $\checkmark$ | $\checkmark$ | $\checkmark$ |  | $\checkmark$ |  |  10.18  |  0.016  |  0.141  |
> |  5  | $\checkmark$ | $\checkmark$ | $\checkmark$ |  |  | $\checkmark$ |  16.34  |  0.007  |  0.038  |
> |  6  | $\checkmark$ | $\checkmark$ |  | $\checkmark$ | $\checkmark$ | $\checkmark$ |  20.67  |  0.004  |  0.022  |
> |  7  | $\checkmark$ | $\checkmark$ | $\checkmark$ |  | $\checkmark$ | $\checkmark$ |  17.50  |  0.005  |  0.027  |
> |  8  | $\checkmark$ | $\checkmark$ | $\checkmark$ | $\checkmark$ |  | $\checkmark$ |  17.34  |  0.013  |  0.044  |
> |  9  | $\checkmark$ | $\checkmark$ | $\checkmark$ | $\checkmark$ | $\checkmark$ |  |  9.65  |  0.015  |  0.083  |
> |  10 | $\checkmark$ | $\checkmark$ | $\checkmark$ | $\checkmark$ | $\checkmark$ | $\checkmark$ |  15.16  |  0.005  |  0.025  |
>
>
> **Q2-2: What is the optimal mask ratio?**
>
> A2-2: The optimal mask ratio is searched and reported in Sec. B and Tab. II in the Appendix. We observe that using a mask ratio of 0.3 is clearly better than others in terms of both energy prediction and structure metrics.
>
> Tab. II is copied below for your convenience:
>
> | Masking ratio | MAE$_\mathrm{E}$ | MAE$_\mathrm{F}$ |  $\Delta P$  |
> |:-------------:|:----------------:|:----------------:|:------------:|
> |      0.1      |       16.18      |      0.0056      |     0.028    |
> |      0.15     |       15.82      |       0.006      |     0.028    |
> |      0.2      |       16.77      |      0.0057      |     0.029    |
> |      0.3      |   $\bf{15.16}$   |   $\bf{0.0050}$  | $\bf{0.025}$ |
> |      0.5      |       17.73      |      0.0066      |     0.032    |
>
>
> Please feel free to reply to us if you have further questions or need clarification.

---

> ### Author Response · Authors · 2022-11-19
> **Answers to Reviewer bSWh (1)**
>
> We appreciate the reviewer for insightful comments and feedback. We answer to each question below:
>
> **Q1: Experiment on additional tasks and on larger dataset?**
>
> A1: Thank you for this constructive feedback. As suggested, we conduct additional experiments: (1) applying our approach to another dataset, and (2) to other tasks.
>
> For (1), we evaluate our approach on the OC20 dataset, and results are reported in Tab. 1. We compare against a few baselines using scores reported in Open-Catalyst-Project (https://github.com/Open-Catalyst-Project). Table 1 indicates that our method is competent on both tasks, outperforming all baselines.
>
> For (2), we employ our approach for an additional task, namely, the molecule assembly task in Sec. 4.4. In addition to the structure optimization conducted in Sec. 4.3, where we start from (almost) optimal structure and optimize the energy to see if the model can recover the stable structure again, the molecule assembly task makes it further challenging by even breaking one or more bonds in the molecule by moving some functional group far away. This task aims to recover the original stable structure from this completely broken one. As shown in Fig. 3 (in Sec. 4.4), our model successfully recovers the disconnected bonds, which could not be achieved by other models. (Note that baseline models fail even with much smaller perturbations in Sec. 4.3.)
>
> Tab. 1 is copied below for your convenience:
>
> | Dataset (Task)  |  QM9  |  |  |  OC20 (S2EF)  |  |  OC20 (IS2RE) |
> |-----------------------------------------|:-------------------------------:|:-------------------------------:|:-------------------------:|:-------------------------------:|:-------------------------------:|:-------------------------:|
> | Model  | MAE$_\mathrm{E}$ ($\downarrow$) | MAE$_\mathrm{F}$ ($\downarrow$) | $\Delta P$ ($\downarrow$) | MAE$_\mathrm{F}$ ($\downarrow$) | MAE$_\mathrm{F}$ ($\downarrow$) | $\Delta P$ ($\downarrow$) |
> | SchNet   |  14.00  |  2.64  |  0.47  |  0.0743  |  1.059  |  0.60  |
> | CGCNN  |  -  |  -  |  -  |  0.0673  |  0.988  |  0.58  |
> | MXMNet   |  5.90  |  1.83  |  1.57  |  -  |  -  |  -  |
> | DimeNet  |  8.02  |  1.79  |  0.58  |  0.0693  |  1.012  |  0.55  |
> | ForceNet  |  18.62  |  0.41  |  0.21  |  -  |  -  |  -  |
> | TorchMDNet (ET)  |  6.15  |  1.15  |  0.32  |  -  |  -  |  -  |
> | GemNet-dT  |  -  |  -  |  -  |  0.0257*  |  -  |  0.18  |
> | SpinConv  |  12.00  |  -  |  -  |  0.0329*  |  -  |  0.21  |
> | Ours ($\mathcal{L}_\text{energy}$ only) |  8.35  |  1.28  |  1.23  |  -  |  -  |  -  |
> | Ours (full model)  |  15.16  |  0.005  |  0.025  |  0.0549  |  0.887  |  0.10  |

---

### Official Review · Reviewer_jtDm · 2022-10-24

**Confidence:** 3
**Correctness:** 3
**Technical Novelty And Significance:** 3
**Empirical Novelty And Significance:** 3
**Recommendation:** 6

**Clarity, Quality, Novelty And Reproducibility:**

The paper writing is clear and the point of view is novel enough. There is sufficient information for reproduction.

**Strength And Weaknesses:**

Strength:

The physical-driven model outperforms the other base line models in the realistic structure optimization task, and the output structures are more physical than other models.

The loss terms are physical meaningful and effective, which shed light on future research.

Weaknesses:

The authors didn’t compare the mask, force and bound terms separately, to see which is the most relevant in terms of the performance improvement.

**Summary Of The Paper:**

In this paper, the authors proposed a physical-driven model and regularization scheme to predict energy for molecular systems. Compared to other baseline models, their full model preforms much better on force prediction, while the energy MAE is slightly lagging behind the state-of-the art MXMNet model. When applying to the structure optimization task, the full model outperforms the other base line models, which demonstrates that the physical-driven model captures the basic physical information instead of simply over-optimizing on the energy estimation. Besides, inspired by the Masked Language Modeling, the authors proposed Masked Atomic Modeling to help the model to discover chemical restrictions.

**Summary Of The Review:**

In summary, the paper proposed a physical-driven model, which performs better on molecular structure optimization and force prediction than the state-of-the-art models. The model may capture the true underlying physics rather than simply fitting. The ideas in the paper would pave the way to more accurate and general physical-driven models in the future research. For these reasons, I recommend receiving this article.

---

> ### Author Response · Authors · 2022-11-19
> **Answers to Reviewer jtDm**
>
> **Q1: Compare the mask, force and bound terms separately?**
>
> A1: Thank you for pointing this out. We refresh the ablation study in Tab. 2, by adding each component separately starting from our base+equation model (#3, #4, #5), and by eliminating each component from the full model (#6 - #10). We use a 6-layer model for this ablation.
> * Mask plays its role in improving the energy estimation. Comparing #7 and #10, having Mask helps the model to improve $\text{MAE}_\text{E}$ without affecting $\text{MAE}_\text{F}$ or $\Delta P$. Looking at #3, solely with Mask, it achieves a nice $\text{MAE}_\text{E}$, but its structure looks suboptimal implied by inferior $\text{MAE}_\text{F}$ and $\Delta P$. By masking parts of the molecule, the model is forced to learn which atoms to be placed in the masked position. This seems to help the model to predict energy better, but solely with this it may not be aware of the entire structure.
> * Bound condition is the most important component for understanding the overall structure. Without it (#9), $\Delta P$ gets significantly worse compared to the full model (#10), while $\text{MAE}_\text{E}$ gets (probably illegally) better by focusing more on the energy itself like baseline models. With the bound condition only (#5), it achieves reasonable $\text{MAE}_\text{F}$ and $\Delta P$, which is not possible with using only Mask (#3) or Force(#4).
> * Force affects all three metrics slightly at the same time. Without Force (#8), all three metrics get slightly worse compared to the full model (#10). With the Force only (#4), however, the $\Delta P$ is suboptimal. We can conclude that the Bound condition is also needed to get a reasonable range of $\Delta P$.
>
> From this observation, we conclude that Mask, Force, and Bound play their own role to improve different metrics.
>
> Tab. 2 is copied below for your convenience:
>
> | No. |  Base  |  [CLS]  |  LJP  |  Mask  |  Force  |  Bound  | MAE$_\mathrm{E}$ ($\downarrow$) | MAE$_\mathrm{F}$ ($\downarrow$) | $\Delta P$  ($\downarrow$) |
> |:---:|:------------:|:------------:|:------------:|:------------:|:------------:|:------------:|:-------------------------------:|:-------------------------------:|:--------------------------------------------:|
> |  1  | $\checkmark$ |  |  |  |  |  |  11.83  |  0.77  |  1.76  |
> |  2  | $\checkmark$ | $\checkmark$ |  |  |  |  |  9.03  |  0.90  |  1.11  |
> |  3  | $\checkmark$ | $\checkmark$ | $\checkmark$ | $\checkmark$ |  |  |  9.70  |  1.91  |  0.814  |
> |  4  | $\checkmark$ | $\checkmark$ | $\checkmark$ |  | $\checkmark$ |  |  10.18  |  0.016  |  0.141  |
> |  5  | $\checkmark$ | $\checkmark$ | $\checkmark$ |  |  | $\checkmark$ |  16.34  |  0.007  |  0.038  |
> |  6  | $\checkmark$ | $\checkmark$ |  | $\checkmark$ | $\checkmark$ | $\checkmark$ |  20.67  |  0.004  |  0.022  |
> |  7  | $\checkmark$ | $\checkmark$ | $\checkmark$ |  | $\checkmark$ | $\checkmark$ |  17.50  |  0.005  |  0.027  |
> |  8  | $\checkmark$ | $\checkmark$ | $\checkmark$ | $\checkmark$ |  | $\checkmark$ |  17.34  |  0.013  |  0.044  |
> |  9  | $\checkmark$ | $\checkmark$ | $\checkmark$ | $\checkmark$ | $\checkmark$ |  |  9.65  |  0.015  |  0.083  |
> |  10 | $\checkmark$ | $\checkmark$ | $\checkmark$ | $\checkmark$ | $\checkmark$ | $\checkmark$ |  15.16  |  0.005  |  0.025  |
>
> Please feel free to reply to us if you have further questions or need clarification.

---

> > ### Comment · Reviewer_jtDm · 2022-12-10
> > **Thank you for the thorough response**
> >
> > I really appreciate your effort giving a response to my review. I have carefully read it as well as the other reviews and responses. I think this work contains some contributions that deserve to be published, so I've updated the score.

---

### Official Review · Reviewer_wZ2H · 2022-10-26

**Confidence:** 3
**Correctness:** 4
**Technical Novelty And Significance:** 3
**Empirical Novelty And Significance:** 3
**Recommendation:** 5

**Clarity, Quality, Novelty And Reproducibility:**

- The writing is overall clear and of high quality.
- The paper is novel in method, but the idea of using force-field relationship in prediction is not quite novel.
- For reproducibility, the code was not included in the submission. The (optional) reproducibility statement was not included in the paper.

**Strength And Weaknesses:**

Strength:
1. Effective use of physics inductive bias. The force-field relationship is crucial for accurate quantum energy understanding.
2. Gives a good energy landscape in terms of molecular structure space. It seems similar in principle to molecular dynamics and helps understand why this method work.

Weakeness:
1. I'm interested to know why the result in Table 1 shows that the force prediction is more accurate yet the energy prediction is not. I wonder if the LJP model gives too much constraint on the energy prediction and affects performance.
2. The experiments are not complete enough:
  - Only performance on QM9 is shown, which is ok since its the standard benchmark in this field. Other datasets (e.g. OC20 ) could also be used as QM9 is often criticized for its non-equilibrium structures of molecules.
  - Strong baselines are missing. ForceNet[1] also learns force in addition to energy and reaches similar resolution to this method in terms of
force prediction.

[1] Hu, W., Shuaibi, M., Das, A., Goyal, S., Sriram, A., Leskovec, J., ... & Zitnick, C. L. (2021). Forcenet: A graph neural network for large-scale quantum calculations. arXiv preprint arXiv:2103.01436.

**Summary Of The Paper:**

The authors propose a transformer model that is trained to learn the force-field relationship of molecules. It has the ability to predict molecular energy without expensive molecular dynamics computation. To achieve this, they use a transformer with a molecular attention block to predict atom force and bond force and then calculate the energy.  A physics-driven parametric energy model, masked atomic modeling, and physics-inspired constraints are combined to improve energy prediction accuracy. On QM9, the model surpasses previous models in terms of force prediction accuracy and structural stability prediction.

**Summary Of The Review:**

The paper is overall clear and well-written. The physics inductive bias seems to be effective. However, the experimental results are not strong in terms of energy prediction. Therefore, I recommend it as borderline reject and I am willing to increase the score if the authors could clarify why the force accuracy is better than energy prediction.

---

> ### Author Response · Authors · 2022-11-19
> **Answers to Reviewer wZ2H (2)**
>
> **Q2-1: Results on another dataset (OC20)?**
>
> A2-1: Thanks for this constructive suggestion. As recommended, we conducted additional experiments on the proposed dataset, OC20, on both tasks, S2EF and IS2RE. The result is added to Tab. 1, where we compare against a few baselines using scores reported in Open-Catalyst-Project (https://github.com/Open-Catalyst-Project). This table indicates that our method is competent on both tasks, outperforming all baselines.
>
> **Q2-2: Comparison with ForceNet?**
>
> A2-2: As suggested, we additionally compare the ForceNet on QM9. The result is also added to Tab. 1. ForceNet achieves the best $\text{MAE}_\text{F}$ and $\Delta P$ among the baselines, indicating its PES is relatively more valid than other baselines. Our model, however, outperforms ForceNet on all three metrics, implying that our model further improves the validity of PES as well as energy prediction under the valid condition. Fig. 2(a) also illustrates that the structure recovered by ForceNet is less perfect than ours.
>
> Please feel free to reply to us if you have further questions or need clarification.

---

> ### Author Response · Authors · 2022-11-19
> **Answers to Reviewer wZ2H (1)**
>
> We appreciate the reviewer for insightful comments and feedback. We answer to each question below:
>
> **Q1-1: Why does Table 1 show more accurate force prediction yet worse energy prediction?**
>
> A1-1: Yes, at a glance, it looks our proposed model is stronger in force prediction but slightly weaker in energy estimation. However, our experiment reveals that previous works only predicted the energy accurately but are not able to generate a legitimate potential energy surface (PES). This is a result of *over-optimization without proper constraints on underlying physics*, e.g., valid molecular structure. This is supported by their significantly higher MAE_F ($\text{MAE}_\text{F}$) and measured structure distortion ($\Delta P$) in Tab. 1, and broken molecular structures they converged, illustrated in Fig. 2 and III in Appendix. These are not physically feasible and incorrect. The energy based on this invalid PES is just simulated to be slightly closer to the GT, but their solution is not within valid physical conditions. In contrast, our model preserves the actual structure, indicating the energy difference we achieve is the actual, valid state-of-the-art.
>
> Tab. 1 is copied for your convenience below:
>
> | Dataset (Task)  |  QM9  |  |  |  OC20 (S2EF)  |  |  OC20 (IS2RE) |
> |-----------------------------------------|:-------------------------------:|:-------------------------------:|:-------------------------:|:-------------------------------:|:-------------------------------:|:-------------------------:|
> | Model  | MAE$_\mathrm{E}$ ($\downarrow$) | MAE$_\mathrm{F}$ ($\downarrow$) | $\Delta P$ ($\downarrow$) | MAE$_\mathrm{F}$ ($\downarrow$) | MAE$_\mathrm{F}$ ($\downarrow$) | $\Delta P$ ($\downarrow$) |
> | SchNet   |  14.00  |  2.64  |  0.47  |  0.0743  |  1.059  |  0.60  |
> | CGCNN  |  -  |  -  |  -  |  0.0673  |  0.988  |  0.58  |
> | MXMNet   |  5.90  |  1.83  |  1.57  |  -  |  -  |  -  |
> | DimeNet  |  8.02  |  1.79  |  0.58  |  0.0693  |  1.012  |  0.55  |
> | ForceNet  |  18.62  |  0.41  |  0.21  |  -  |  -  |  -  |
> | TorchMDNet (ET)  |  6.15  |  1.15  |  0.32  |  -  |  -  |  -  |
> | GemNet-dT  |  -  |  -  |  -  |  0.0257*  |  -  |  0.18  |
> | SpinConv  |  12.00  |  -  |  -  |  0.0329*  |  -  |  0.21  |
> | Ours ($\mathcal{L}_\text{energy}$ only) |  8.35  |  1.28  |  1.23  |  -  |  -  |  -  |
> | Ours (full model)  |  15.16  |  0.005  |  0.025  |  0.0549  |  0.887  |  0.10  |
>
> **Q1-2: Does LJP model give too much constraint on the energy prediction and affects performance?**
>
> A1-2: Regarding the concern about rigidity of LJP, our new ablation study (Tab. 2) indicates that having LJP improves $\text{MAE}_\text{E}$ by ~5 meV (from 20.67 to 15.16) without significantly sacrificing $\text{MAE}_\text{F}$ (from 0.004 to 0.005) and $\Delta P$ (from 0.022 to 0.025), comparing #6 (the one with everything but LJP) and the full model (#10). Note that our model also can achieve better $\text{MAE}_\text{E}$ by ignoring the physical constraints (#1, #2, #3 in Tab. 2), but we do not claim this as our best result since its $\text{MAE}_\text{F}$ and $\Delta P$ are significantly higher, beyond the range considered correct. From this, we conclude that LJP does help the model to predict energy more accurately, rather than giving too much constraint.
>
> Tab. 2 is copied here:
>
> | No. |  Base  |  [CLS]  |  LJP  |  Mask  |  Force  |  Bound  | MAE$_\mathrm{E}$ ($\downarrow$) | MAE$_\mathrm{F}$ ($\downarrow$) | $\Delta P$  ($\downarrow$) |
> |:---:|:------------:|:------------:|:------------:|:------------:|:------------:|:------------:|:-------------------------------:|:-------------------------------:|:--------------------------------------------:|
> |  1  | $\checkmark$ |  |  |  |  |  |  11.83  |  0.77  |  1.76  |
> |  2  | $\checkmark$ | $\checkmark$ |  |  |  |  |  9.03  |  0.90  |  1.11  |
> |  3  | $\checkmark$ | $\checkmark$ | $\checkmark$ | $\checkmark$ |  |  |  9.70  |  1.91  |  0.814  |
> |  4  | $\checkmark$ | $\checkmark$ | $\checkmark$ |  | $\checkmark$ |  |  10.18  |  0.016  |  0.141  |
> |  5  | $\checkmark$ | $\checkmark$ | $\checkmark$ |  |  | $\checkmark$ |  16.34  |  0.007  |  0.038  |
> |  6  | $\checkmark$ | $\checkmark$ |  | $\checkmark$ | $\checkmark$ | $\checkmark$ |  20.67  |  0.004  |  0.022  |
> |  7  | $\checkmark$ | $\checkmark$ | $\checkmark$ |  | $\checkmark$ | $\checkmark$ |  17.50  |  0.005  |  0.027  |
> |  8  | $\checkmark$ | $\checkmark$ | $\checkmark$ | $\checkmark$ |  | $\checkmark$ |  17.34  |  0.013  |  0.044  |
> |  9  | $\checkmark$ | $\checkmark$ | $\checkmark$ | $\checkmark$ | $\checkmark$ |  |  9.65  |  0.015  |  0.083  |
> |  10 | $\checkmark$ | $\checkmark$ | $\checkmark$ | $\checkmark$ | $\checkmark$ | $\checkmark$ |  15.16  |  0.005  |  0.025  |

---

### Author Response · Authors · 2022-11-19
**Summary of revision**

We sincerely appreciate constructive feedback from all reviewers. Accordingly, we updated our paper as follows:

* An additional dataset (OC20) is added to our experiment. (Sec. 4.1, 4.2)
* An additional task (Molecule assembly) is added to our experiment. (Sec. 4.4)
* Refreshed our ablation study to more clearly show effect of each idea. (Sec. 4.5)
* An additional ablation study (masking ratio for MAM) is added. (Sec. B in Appendix)
* Added 102 examples of structure optimization by our model and competing models. (Sec. D in Appendix)

In the manuscript, the edited parts are marked in blue fonts.

We are welcome additional deeper discussion. Please feel free to reply if you need clarification or have further questions.

---

### Decision · Program_Chairs · 2023-01-20

**Decision:**

Reject

**Justification For Why Not Higher Score:**

I think I just described this in the meta review above...

**Justification For Why Not Lower Score:**

N/A

**Metareview: Summary, Strengths And Weaknesses:**

All reviewers raised several points of criticism, such as missing novelty of concepts,  unclear ablation studies and not entirely convincing experimental validation. Even after the rebuttal and discussion phase, these concerns largely remained and all scores were in the negative range. I mostly agree with this over-all perception, and therefore I vote for rejection of this paper.